# Electrochemical Biosensors as Potential Diagnostic Devices for Autoimmune Diseases

**DOI:** 10.3390/bios9010038

**Published:** 2019-03-04

**Authors:** Anca Florea, Gheorghe Melinte, Ioan Simon, Cecilia Cristea

**Affiliations:** 1Analytical Chemistry Department, Iuliu Haţieganu University of Medicine and Pharmacy, 400349 Cluj-Napoca, Romania; florea.ancas@yahoo.com (A.F.); Melinte.Gheorghe@umfcluj.ro (G.M.); 2Department of Surgery IV, CFR Hospital, Iuliu Haţieganu University of Medicine and Pharmacy, 18 Republicii St, 400015 Cluj-Napoca, Romania; isimon@umfcluj.ro

**Keywords:** immunosensors, biomarkers, autoimmune diseases

## Abstract

An important class of biosensors is immunosensors, affinity biosensors that are based on the specific interaction between antibodies and antigens. They are classified in four classes based on the type of employed transducer: electrochemical, optical, microgravimetric, and thermometric and depending on the type of recognition elements, antibodies, aptamers, microRNAs and recently peptides are integrating parts. Those analytical devices are able to detect peptides, antibodies and proteins in various sample matrices, without many steps of sample pretreatment. Their high sensitivity, low cost and the easy integration in point of care devices assuring portability are attracting features that justify the increasing interest in their development. The use of nanomaterials, simultaneous multianalyte detection and integration on platforms to form point-of-care devices are promising tools that can be used in clinical analysis for early diagnosis and therapy monitoring in several pathologies. Taking into account the growing incidence of autoimmune disease and the importance of early diagnosis, electrochemical biosensors could represent a viable alternative to currently used diagnosis methods. Some relevant examples of electrochemical assays for autoimmune disease diagnosis developed in the last several years based on antigens, antibodies and peptides as receptors were gathered and will be discussed further.

## 1. Introduction

The body immune system consists of a complex network of cells and molecules working together to protect the body against various diseases. However, under certain circumstances, the immune system could attack and damage body’s own tissues, organs and cells, resulting in autoimmune diseases (ADs) [1]. There has been a significant increase in autoimmune diseases (ADs) diagnosed over the last decade, caused by genetics or environmental factors, like lifestyle choices or pollution [2]. Unfortunately, the symptoms of ADs are extremely vast, implying difficulties for physicians to establish a diagnosis. Most importantly, the prevention and the management of ADs are difficult, which affects the life quality and the life expectancy of the patients. The most frequent ADs include celiac disease, rheumatoid arthritis, systemic lupus erythematosus, multiple sclerosis, and psoriasis.

It is estimated that ADs affect 5–10% of the general population. Only in the USA, the National Institute of Health estimate that up to 23.5 million people suffer from ADs (for comparison, cancer affects up to 9 million and heart disease up to 22 million) and that the prevalence is rising [3]. About 10% of the adult population in Europe was diagnosed with an AD only in 2018. Unfortunately, in the last several decades, significant changes occurred in Western dietary habits, in the quality of the environmental surroundings and in the degree of exposure to pollution and infectious diseases. Together with an increased stress load, these changes caused an increase in ADs in Western countries [4]. Through their chronic and debilitating nature, ADs are becoming a massive burden on patients, their families, and society, leading to high medical costs and reduced quality of life. The etiology of most ADs remains unknown and there is currently no cure, thus the diagnosis in early stages of the disease is of paramount importance for a positive outcome for the patients. Currently, ADs are diagnosed based on physician assessment combined with core laboratory tests. However, these tests are not sensitive enough to detect early molecular events. Too often, the disease is diagnosed too late, in a stage when the tissue damage is irreversible and the symptoms are hard to control [5]. For instance, rheumatoid arthritis (RA) is a common chronic AD that, if left untreated, results in severe joint destruction leading to impaired physical function and work disability. Irreversible damage frequently occurs early in RA. The early diagnosis of RA is essential to avoid an aggressive treatment and to prevent joint damage and disability, therefore there is an urgent need to diagnose RA as early as possible. Lupus is another AD that can affect many different body systems—including joints, skin, kidneys, blood cells, brain, heart and lungs.

In the last several years, personalized medicine received much attention due to the possibility of establishing an accurate diagnosis even before any symptom of the disease appears, which is fundamental for patient survival. The main challenge in medical diagnosis is the early diagnosis and personalized care of patients using non-invasive methods. To reach this goal, reliable methods to detect and monitor specific biomarkers that indicate a pathological event are required. The demands of new analytical methods that are reliable, cost-effective, accurate and capable of optimizing the existing protocols by making them faster and more economical has impressively grown. In this light, electrochemical biosensors are a very attractive alternative to other analytical devices, providing multiplexed analysis, fast response, sensitivity, specificity and lower costs [6] compared with validated analytical methods.

Electrochemical biosensors based on antibodies (Abs), antigens (Ags) and peptides as recognition elements have been the most investigated due their high affinity, versatility and commercial availability of the biological elements. The huge interest towards them is justified by their use as high-throughput laboratory instruments and as small devices for point-of-care (POC) analysis suitable for onsite analysis.

This paper reviews the biomarkers and electrochemical biosensors reported in literature in the last five years for the detection of AD biomarkers. The first part deals with the description of general aspects in the design and development of electrochemical platforms for the detection of ADs biomarkers, while the second part briefly describes common biomarkers and recent progress made for their detection using biosensors based on Abs/Ags and peptides.

## 2. Platform Design for AD Biosensors

Biosensors developed for the diagnosis and detection of ADs are mostly affinity biosensors which have as a working principle the specific antigen (Ag)–antibody (Ab) interaction combined with different transducers. These are known as immunosensors. The signal related to formation of the immunocomplex is usually generated by the use of various labels. Label free detection is also possible for the quantification of the immune-complex thanks to the modern transducer technology [7]. The working principle of immunosensors is based on specific recognition of Ags by Abs that link with high binding constants [8].

Most of the immunosensors reported for ADs are based on labeled methods and a direct and indirect format. They usually employ an Ag as recognition element since the targets in ADs are usually autoantibodies. Ags are firstly immobilized on the electrode surface. Then, the analyte is added (the sample containing autoantibodies present in ADs). The Ags selectively recognize and bind the Abs from the sample, and the formation of the Ag–Ab complex can be assessed using secondary Abs, which are usually labeled with an enzyme. Upon the addition of the enzyme’s substrate, the product of the enzymatic reaction (electrochemically active) enables the quantification of the target analyte. In the cases where the target analyte is an antigen, for example an interleukin (IL), a sandwich assay may be employed, using a primary Ab as recognition element.

In sandwich assays after the recognition of the Ags by the Abs immobilized on the surface of the electrodes, a second labeled Ab binds to a second binding site of the Ag. The detection in this case needs a label such as enzymes (with its specific substrate) or nanoparticles which could provide the redox signal.

Even though the concept of directly or indirectly detecting the binding event is quite simple and elegant, the development of such a device demands a multidisciplinary approach, combining the expertise of immunologists, biochemists, engineers, and materials scientists. The main challenge remains in developing the intimate interface between the biologic component and the transducer.

Although progress has been made to obtain new synthetic molecular receptors in the laboratory such as phages, molecular imprinted polymers [9,10,11] and aptamers [12,13], Abs and Ags [14] are still the most used bioreceptors among biosensors in clinical analysis and medical diagnosis, including in those for ADs. Lately, peptides have been more and more used in the role of molecular receptors. Their biocompatibility and their structural similarity with proteins made them an ideal substitute for proteins as a recognition element for different AD biomarkers. The most known methods for peptide immobilization on an electrode surface are by covalent bonds or by using self-assembled monolayers (SAMs). Once immobilized, peptides can play different roles, like bioreceptors or enzymatic substrate [15].

### 2.1. Bioelement Immobilization Methods

The immobilization method employed to link the bioelement to the transducer is of paramount importance for the performance of the biosensor. For example, the bioelement can be directly adsorbed onto the electrode, or can be linked via SAMs. Adsorption of the bioelement is a simple procedure, however, it does not allow a controlled orientation of the recognition element for the proper binding of the Ag or the Ab. Additionally, fouling of the electrode surface may occur. Thus, methods that allow a controlled, oriented immobilization of the bioelement are preferred. For example, SAMs can be readily formed onto gold electrodes via Au-S linking. Thiolated Abs can be employed, or a SAM layer may be used to link Abs via carboxyl or amino groups. Polymers exhibiting various functional groups, such as poly (sodium-4-styrensulfonic acid) [16] or polydopamine [17] are also employed for oriented immobilization.

Important progress has been made in the last years in the development and design of immunosensors regarding the immobilization method in particular in the use of nanomaterials [18].

Integrating nanomaterials into immunosensors have the final goal of obtaining nanostructured surfaces for the immobilization of the Ab or the Ag and enhance the performance of the biosensors. Nanomaterials, such as carbon nanotubes, graphene or metallic particles have high conductivity and electrocatalytic effect, improving the electron transfer and giving higher signals. The roughened surface generated by the modification of the electrode with nanomaterials facilitates the attachment of a higher number of biomolecules to the electrode surface leading to higher sensitivity [19]. Most nanomaterials are biocompatible [20]. Other immobilization strategies may be considered and applied for AD immunosensors, given the fact that they were successfully applied for other targets. Although not reported so far for AD biosensors, magnetic beads functionalized with various groups, such as avidin, protein A or G, can also be used as support for the immobilization of the Ab or the Ag with the advantage of high loading capacity of biomolecules, easy separation and easy washing steps [21]. Another immobilization strategy is based on using aryl diazonium salts to link the Ab directly onto the electrode by electroaddressing [22] or using polymer composites to covalently link the capture Ab in an oriented manner [23]. These strategies could be easily adapted and employed for the immobilization of bioelements specific for AD biomarkers. Peptides could be considered such bioelements, having a low cost, fast and easy synthesis and the capacity of specific cleavage in order to specifically recognise AD biomarkers.

### 2.2. Signal Generation

Various strategies have been used to transduce and amplify the signals of Ag–Ab interactions. Most strategies rely on the use of labels such as enzymes (Figure 1), electroactive molecules, metal ions; however, label-free detection by electrochemical impedance spectroscopy (EIS) has been also widely exploited due to its simplicity and possibility for real-time monitoring of various analytes [20].

#### 2.2.1. Labels in Electrochemical Immunosensors

Electrochemistry offers versatile methods for sensing Ag–Ab complex formation that are sensitive and accurate, have a low cost, low sample volume consumption and are easy to integrate in a portable device [24]. Voltammetry (cyclic voltammetry CV, differential pulse DPV and square wave voltammetry SWV) and chronoamperometry are the electrochemical methods most commonly used to convert the Ab–Ag binding event into a measurable signal, the current response. Voltammetry measurements can be performed in quiescent, under solution stirring or in flow that improves the mass transport to the working electrode and lowers the detection limit to reach the range important from the clinical point of view. Moreover, using flow or batch injection analysis speeds up the test time and improves the throughput of samples [25]. Since Ags and Abs are not intrinsically electroactive, a label is usually linked to the Ag or the Ab to produce the electrochemical signal [26]. Usually, a labeled secondary Ab is used to generate the signal, in order to avoid labeling Abs for each specific Ag. This is a generic labeled IgA/G antiAb. The label is usually an enzyme or nanomaterials such as metallic or semiconductor particles. Recently, nanomaterials have been employed as nanocarriers in the fabrication of immunosensors. These include carbon nanotubes (CNTs), metallic particles, dendrimers, silica particles, graphene or magnetic beads. Due to their high surface area, they can load a high number of enzymes or electroactive species, thus leading to signal amplification (additionally to their excellent conducting and electro-catalytic properties) and better detection limits [27]. Additionally, they can be easily conjugated with Abs via covalent linkages if they are functionalized by amine or carboxylic groups [28]. For example, Wang et al. used CNTs as carriers to load a high number of enzymes by covalent bonds, which lead to great signal amplification (100 times compared to single enzyme labeling). The loading was estimated to be 9600 enzymes per CNT. The labeling strategies were applied for an immunoassay based on magnetic beads to detect IgG and a low limit of detection (LOD) of 500 fg mL^−1^ was achieved [29]. Similarly, Rusling’s group loaded horseradish peroxidase (HRP) onto CNTs to label the secondary Ab in an immunosensor for the prostate specific Ag (PSA) reaching a LOD of 4 pg mL^−1^ [30].

Electroactive labels such as metallic nanoparticles, in particular gold and silver, are employed in the construction of immunosensors for ADs. Usually conjugated with the detection Ab, metallic nanoparticles generate signals by their redox properties in certain conditions e.g., in acidic conditions for gold nanoparticles (AuNPs) [31]. For example, Dequire et al. used colloidal gold as labels, the signal being generated by anodic stripping voltammetry after oxidative gold dissolution in acid [32] while Liu et al. directly quantified gold used as label by stripping analysis without dissolution [33]. Moreover, AuNPs can be used as labels with the role of electrocatalyst, favorizing the redox reaction of a redox active compound added to the system. Das et al. employed AuNPs as electrocatalyst for the reduction of *p*-nitrophenol to *p*-aminophenol greatly enhancing the electrochemical signal of the immunosensor [34]. Hybrid silver/gold particles can be employed as labels as well, signal generation being obtained by acidic dissolution of silver and its subsequent stripping analysis [35]. Electrochemical stripping transduction of semiconductor nanoparticles such as CdS, ZnS, CuS or PbS has also been employed to generate signals related to the immuno-binding event.

Enzyme, such as HRP, Glucose oxygenase (GOX) or Alkaline phosphatase (AP) are commonly employed in immunosensors for the electrochemical signal tracing through their biocatalytic reaction [36]. The detection Ab is usually labelled with the enzyme to quantify the captured Ag, in a sandwich format, or the Ab, in indirect assays. Even though the enzyme amplifies the electrochemical responses, it is always necessary to use a mediator either added in the detection solution or immobilized onto the electrode surface to accelerate the electron transfer on the electrode surface [36,37,38]. For example, for HRP hydroquinone or *o*-aminophenol are added in the system to mediate the electron transfer between HRP and its substrate H_2_O_2_ [39,40]; however, reagentless immunosensors based on direct electrochemistry of the enzyme were also developed. In this case, the immobilization method of the enzyme is crucial to have exposed active centers of the enzyme available for electron transfer and most immunosensors work with the direct format of immunoassay [36]. Direct electrochemistry of HRP is based on Fe(III) to Fe (II) conversion [41]. For GOX, the direct electrochemistry is based on the exposure of its FAD cofactor for facilitating direct electron transport [42].

#### 2.2.2. Label-Free Electrochemical Immunosensors

Even though the various labeling strategies presented above are important not just to generate the signal but also to lower the detection limit (e.g., nanocarriers), immunosensors for direct, label-free measurements of various biomarkers are attractive, as they also provide real-time monitoring. EIS is an electrochemical technique, it is widely used as detection method for label free immunosensors. Among the advantages of developing label free immunosensors, it is important to remember the high decreasing of time needed for detection, and the cost-reduction by avoiding the labeling step. To develop a sensor without a label that can generate a high electrochemical signal, the EIS response must by amplified and ways to increase the transfer resistance must be found.

Biomarkers like Myelin basic protein (MBP), or interleukins and tumor necrosis factor alpha (TNF) were used to diagnose ADs, using label free EIS immunosensors. In order to obtain a good sensitivity, the electrodes have been modified with materials like platinum, gold, TiO_2_ or polymers in order to obtain a higher signal, to increase the electroactive surface and to obtain lower limits of detection [43,44,45].

## 3. Biosensors for ADs Based on Antibodies and Antigens

Due to the limited knowledge on the pathogenesis of ADs, the medical treatment is mainly based on treating the symptoms rather than curing the disease. To reduce the severity of the symptoms and the irreversible damage to organs or joints, it is important to detect and treat the disease in early stage. The diagnosis of ADs is usually based on symptoms and laboratory tests confirming the presence of serological and genetic biomarkers, such as autoantibodies or complement proteins [47]. Biomarkers are markers that have a characteristic that can be measured objectively and evaluated as an indicator of physiological or pathogenic processes or pharmacologic response to an intervention [48]. Several biomarkers were identified for ADs [49,50], but more research is needed in this field. Some biomarkers are more sensitive and specific for a certain AD, for example anti-cyclic citrullinated peptide (anti-CCP) Ab for RA, while others are non-specific exhibiting elevated levels in a number of diseases, such as the pro-inflammatory cytokines.

Celiac disease (CD) is one of the most common AD triggered by the ingestion of wheat gluten and similar proteins in barley and rye, which produces and autoimmune response that induces atrophy and hyperplasia in the small intestine [51]. The worldwide prevalence of celiac disease increased from 0.03% in the 1970s to 1% to this day [52]. Diagnosis is usually based on endoscopy with a small biopsy and evaluation of the serological markers [51]. Antigliadin Abs (AGA) and anti-transglutaminase Abs (anti-tTG) are present in the blood and small intestine mucosa of these patients, serving as specific biomarkers of the disease [53,54]. IgA isotypes are the most specific, but are not present in 2–5% patients diagnosed with CD. In this case, IgG isotype is considered [55]. An interesting concept towards point-of-care diagnostics of celiac disease was reported by Gianneto et al. The group developed a portable device in which the electrochemical signal was acquired and processed through a developed IoT-WiFi integrated board that is capable of sharing the results via the cloud with doctors or caregivers. The electrochemical platform consisted of screen-printed electrodes functionalized with AuNPs on which transglutaminase was immobilized to capture anti-tTG Abs. The amperometric signal was generated via a secondary Ab labelled with AP [56]. Simultaneous detection of multiple biomarkers increases the accuracy of the detection. For example, Cosa-Garcia’s group developed an immunosensor capable of detecting two biomarkers, each with two isotypes, for celiac disease IgA AGA, IgB AGA, IgA anti-tTG and IgB anti-tTG. Screen-printed electrodes were modified with nanostructured carbon-metal hybrid and the Ags, AGA and tTG were immobilized onto the electrodes to capture the specific Abs. The detection Ab labeled with AP and a mixture of 3-indoxyl phosphate with silver ions was used as substrate. LODs between 2.45 and 3.16 U mL^−1^ were obtained, which are in the relevant clinical range. The sensor was also tested on real patients’ serum [57].

Multiple sclerosis (MS) is one of the most common ADs and it is characterized by an abnormal inflammation process that leads to myelin destruction and to irreversible changes in the nervous system. MS affects mostly young teenagers born in high developed countries. The golden standard used nowadays for the diagnosis of MS is magnetic resonance imaging (MRI), but, even though the MRI may fail, considering the fact that symptoms are not entirely corelated with the clinical setting. Considering this, the need for new diagnosis methods is even more justified. Patients with MS can develop a series of autoantibodies that are present in the biological fluids, autoantibodies that cannot be found in the biological fluids of healthy subjects, and that can be used as diagnosis and prognosis biomarkers [58].

Studying the myelin oligodendrocyte glycoproteins has be found that [Asn31(Glc)]hMOG(30–50) glycoprotein is able to differentiate the serum of MS positive patients from the serum of MS negative ones, recognizing the autoantibodies developed by MS patients. An antigenic probe, CSF114(Glc), was designed and synthesized in order to detect the minimal epitope of the above glycoprotein, an N-glucosylated amino acid on a type I0 b-turn. CSF114(Glc) was able to differentiate a significant proportion of MS sera from negative MS sera, recognizing IgMs in about 30% of MS patients [59,60].

Rheumatoid arthritis (RA) is another common AD with a prevalence of about 0.5–1%. The main symptom in RA is joint pain and deformation due to synovial chronic inflammation that leads to joint destructions and disability [61]. Clinical remission is possible in early RA; however, it is not achievable in all patients, since RA is a rather heterogeneous disease [62]. Thus, the diagnosis of RA in early stages is crucial for the outcome of the disease. Related to the prognosis of the disease, there are several biomarkers, which point to a poor prognosis, with rapid joint distraction, if a patient is tested seropositive: high acute phase reactants (erythrocyte sedimentation rate, ESR, C reactive protein, CRP), rheumatoid factor (RF) and anti-CCP Ab [63,64]. RF and anti-CPP are common biomarkers used for the clinical detection of rheumatoid arthritis. Anti-CPP is more specific than RF and is produced in the mucosal tissues and at the point of inflammation [65]. The concentration of CPP in serum and synovial fluid show the onset of RA and also points to its severity [66]. Another biomarker for RA is the macrophage migration inhibitory factor (MIF) is present in high concentrations in the blood and synovial liquid of patients suffering from this disease [17]. Other (non-specific) biomarkers for RA include tumor necrosis factor (TNFα), interleukin 6 (IL-6), osteopontin, osteocalcin, amino-terminal telopeptide of type 1 collagen (NTX), carboxyl-terminal telopeptide of type 1 collagen (CTX), matrix Metalloproteinase 3 (MPP)-3 and so on [64].

Other examples of immunosensors for the detection of common biomarkers expressed in ADs are provided in Table 1. Most of the work reported in the literature employs electrochemistry as a detection technique given its high sensitivity. Detection limits in the fg mL^−1^ range of concentrations were reported by electrochemical immunosensors.

## 4. Biosensors for ADs Based on Peptides

Due to their capacity to self-assemble in highly-ordered 1D, 2D and 3D structures, peptides represent one of the most versatile tool in the creation of flexible frameworks. Peptides can be used as bioreceptors for the developing of novel biosensors due to their ability to modify their secondary structure by modifying the amino acid (AA) sequence and to optimize the interactions between adjacent peptides.

Short peptides, up to 10 AA residues, can easily be obtained by synthesis, using simple, short and low-cost techniques, having a good biocompatibility, better chemical and conformational stability then proteins and offering short response time in electrochemical detection [74].

Peptide based electrochemical biosensors developed in recent years were characterized by good sensitivity and the ability for miniaturization, both of them being interesting criteria in biosensors field of research (Table 2). Even though the good sensitivity recommends the use of peptides, there is still work to be done. Most of the published methods did not report platforms characterization using real samples from real patients or tests using multiple analytes that could interfere with the detection, proving lack of selectivity. Therefore, the multiplex and simultaneous analysis of target analytes is necessary. There are also studies to be conducted in order to prove the repeatability, reproducibility and stability of peptide based biosensors [15].

Developing peptide based biosensors instead of immunosensors or aptasensors is a continuously growing tactic, and more and more articles are being published (Figure 2).—from 2008, whenonly eight articles were reported in Scopus [75] having the words *peptide based biosensor* in their title, up until 2018, when 20 articles were published using the same criteria. Starting in 2008 when 34 articles reported peptide biosensors in their title, abstract or keywords, the number increased continuously up until 2018 when 89 articles were published under the same criteria. The ScienceDirect database was used for this classification.

Real-Fernandez et al. demonstrated that the synthetic glucosylated myelin oligodendrocyte glycoprotein fragment, (Asn31(Glc)hMOG(30–50)), was able to detect autoantibodies in MS patients. Moreover, the detection MS autoantibodies was attributed to the N-linked glucosyl moiety. After the optimization of recognition properties, a specific peptide antigenic probe was developed. The next step was to develop a label free serodiagnosis SPR biosensor for MS, based on the specific immobilization of CSF114 on a sensor chip surface, in order to diagnose the MS by the differences between the number and the type of autoantibodies detected in MS patients’ sera, and the ones detected in healthy individuals The differences between the MS and healthy individuals mean values were 94.6 vs. 48.9 Response Units, respectively. The results obtained with the biosensor were similar to those obtained with an already validated ELISA [58].

Using a gold electrode as immobilization surface, the result was enhanced and a detection limit was obtained, with a calibration curve for the anti-CSF114 Abs in the range of 1.25–20 µg mL^−1^, allowing using the biosensor even for MS prognosis [60]. Furthermore, after labeling CSF114 with a ferrocenyl moiety, the peptide was immobilized on a platinum electrode, without any previous surface modification. The developed biosensor was characterized using CV, the interactions between Fc-CSF114(Glc) and autoantibodies being characterized by a shift of the oxidation potential with several tens of millivolts towards more positive values using autoantibodies concentrations higher than 0.1 mg mL^−1^ [61].

Matrix Metalloproteinases (MMPs) are a large group of Calcium-dependent endopeptidases. Their overexpression is usually correlated with a series of pathological conditions like inflammatory diseases or cancer. Moreover, MMPs are between the most used biomarkers in electrochemical peptide based detection. A number of biosensors for the selective detection of MMPs are reported in the literature (Table 2) using different electrode architectures and obtaining different results. The best limit of detection was obtained for MMP-7 (5 × 10^−5^ ng mL^−1^) [76] using a peptide and single-stranded DNA S_1_ modified platinum nanoparticles immobilized on AuNP modified Glassy Carbon Elecrodes (GCEs). In this case, the peptide served as a recognition element for the MMP-7. After the recognition event, the PtNP-S_1_ bioconjugates were released from the electrode surface. The indirect detection was made by measuring 4-chloro-1-naphthol oxidation using DPV, after a previous hybridization of the remaining S_1_ the electrode surface, after the MMP-7 recognition and the formations of DNA nanoladders, ideal nanocarriers for the loading of the enzyme needed for 4-chloro-1-naphthol oxidation (Figure 3). The biosensor presented a linear range for MMP-7 detection, using DPV measurements of 2 × 10^−4^–20 ng mL^−1^ [76].

A simpler electrochemical peptide based biosensor for the selective detection of MMPs was developed by Donk-Sik Shin et al. In this case, a methylene blue modified peptide was immobilized on a 300 µM gold electrode. MMP-9 interaction with the platform leads to peptide cleavage and to a loss in the SWV signal (Figure 4). The biosensor is characterized by a LOD of 5.52 ng mL^−1^ and a linear range of 5.52–4.6 × 10^3^ ng mL^−1^. Even though its analytical parameters are not as good as in the above study, this biosensor has the advantage that it was not only tested on real samples like serum samples, but it can also detect live MMP-9 release from monocytes [77].

## 5. Conclusions

ADs represent nowadays an important health issues, taking into account that annually new cases are reported and, for most of this diseases, no cure is available on the market. Therefore, an early diagnosis will help practitioners and also patients, reducing the incidence of premature death. Due to the advancements in molecular biology and immunology, almost every AD possesses its own set of biomarkers (proteins, Abs or peptides), which could be found in detectable concentrations in body fluids. The main challenge remains the implementation of point of care devices able to detect the ADs’ biomarkers before any symptoms appear. Many attempts are already reported in the literature and relevant reviews were dedicated to electrochemical biosensors, which represent viable alternatives for the development of point of care devices.

Several examples of electrochemical biosensors reported in the recent years for the detection of relevant biomarkers in ADs diagnosis were reported. The attention focused on: different approaches of bioelements’ immobilization, integration of nanomaterials for improving the sensitivity, multianalyte detection and on the type of analyzed biological samples. Taking into account that electrochemical immunosensors could detect biomarkers (proteins and peptides) in fM ranges without any laborious pre-treatment of the real samples, the integration on these sensors into decentralized analyzers will be the next logical step. In this vast pathology with so many different manifestations, early diagnosis will lead to the improvement of the patient life quality and health care cost reductions.

However, there are gaps to be filled up from the bench to the market, but with the last achievement in nanomaterials technology and molecular biotechnologies, electrochemical biosensors could have a bright future as a potential diagnostic devices for ADs.

## Figures and Tables

**Figure 1 biosensors-09-00038-f001:**
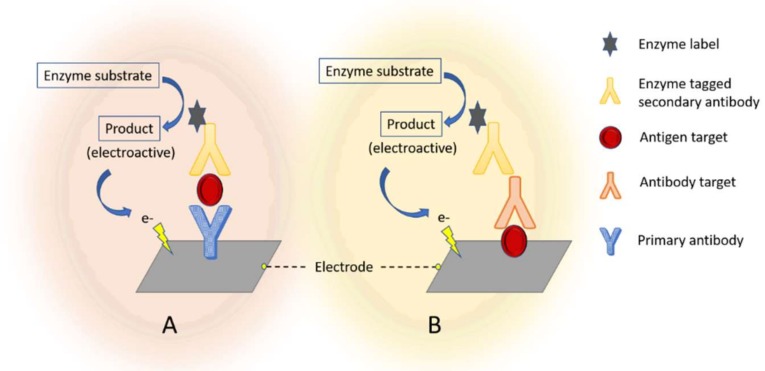
Scheme showing the basic components of electrochemical immunosensors with enzyme labels: A. antigen’ detection principle; B antibody’s detection principle (adapted from [46]).

**Figure 2 biosensors-09-00038-f002:**
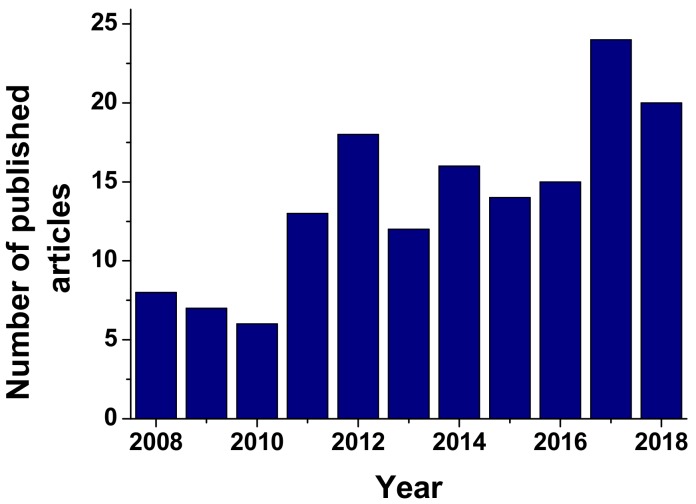
Number of articles containing *Peptide biosensor* in their title published in Scopus [75].

**Figure 3 biosensors-09-00038-f003:**
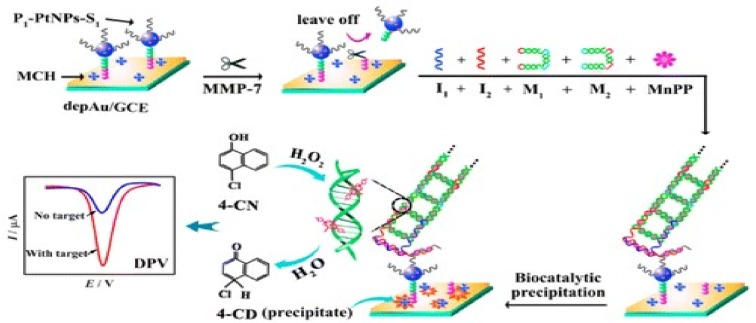
Schematic illustration of matrix metalloproteinases 7 (MMP-7) electrochemical biosensors [76].

**Figure 4 biosensors-09-00038-f004:**
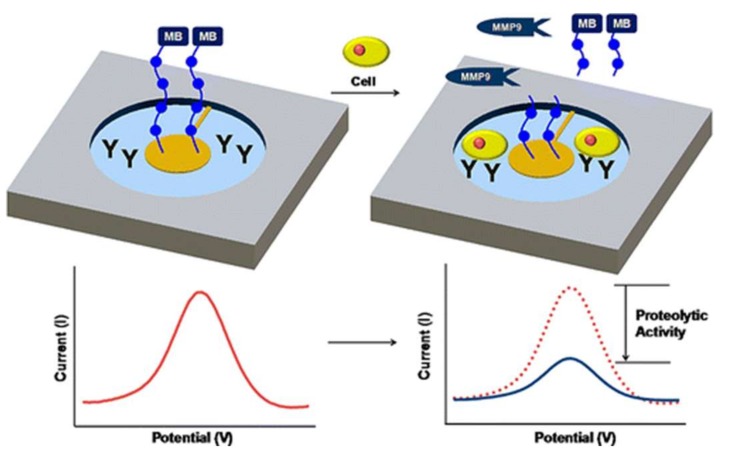
Schematic illustration of matrix metalloproteinase MMP-9 electrochemical biosensors [77].

**Table 1 biosensors-09-00038-t001:** Immunosensors based on antibodies for diagnosis and monitoring of autoimmune diseases.

Target	Electrode Architecture	Type of Assay	Label	Detection Method	LOD	LR	Sample	Ref.
**Celiac Disease**								
IgA anti-tTG IgB anti-tTG	AuNPs/SAM-GCE	Indirect	AP	DPV	3.2 AU mL^−1^ 1.4 AU mL^−1^	0–30 AU mL^−1^	Real patients serum	[55]
	CNTs/AuNPs-SPE	Indirect	AP	CV	9.1 U mL^−1^ 9.0 U mL^−1^	-	Real patients serum	[66]
	Au/SAM-GCE	Indirect	HRP	CV	1.7 AU mL^−1^ 2.7 AU mL^−1^	0–30 AU mL^−1^	Serum	[67]
IgA anti-tTG	Graphite epoxy	Indirect	HRP	Chronoamperometry	-	-	Real patients serum	[68]
	GQD/AuNPs/MWCNTS/PAMAM	Direct	-	DPV with redox probe	20 fg mL^−1^	-	Spiked serum	[69]
Anti-tTG	poly (sodium-4-styrensulfonic acid)- gold SPE	Indirect	POD	EIS	-	-	Real patients serum	[16]
	Multichannel SPE array	Indirect	CdSe QDs	DPV	7 U mL^−1^	0–40 U mL^−1^	Spiked serum	[70]
AGA	Gold electrodes with carboxylic-ended bipodal alkanethiol	Indirect	HRP	Chronoamperometry	46 ng mL^−1^	0–1 μg mL^−1^	Real patients serum	[53]
**Rheumatoid Arthritis**								
MIF	AuNPs-NTiP-Thi-gold electrode	Direct	-	DPV with redox probe	0.7 ng mL^−1^	0.03–230 ng mL^−1^	Real patients serum	[67]
**Multiple Sclerosis**								
Anti-MBP	Gelatin-NTiP-Pt electrodes	Direct	-	EIS	0.15 ng mL^−1^	0.48–2500 ng mL^−1^	Spiked serum Spiked CSF	[42]
**Non-Specific Biomarkers**								
HIgG	GO-SPE	Direct	-	CV with redox probe	1.70 ng mL^−1^	2–100 ng mL^−1^	Urine	[71]
	AuNPs-PDA-GO	Sandwich	AgNPs/carbon nanocomposite/benzoquinone	DPV	0.001 ng mL^−1^	0.1–100 ng mL^−1^	Spiked serum	[17]
IL-17	Graphene-GC	Sandwich	cadmium-polystyrene beads	SWV	50 fg mL^−1^	0.1 pg ^−1^ ng mL^−1^	Spiked serum	[72]
IL-12	Electroplating gold onto a disposable printed circuit board electrode	Direct	-	EIS	<100 fM	0–25 pg mL^−1^	Spiked serum	[43]
TNFα	Poly(3-thiophene acetic acid)-ITO	Direct	-	EIS	3.7 fg mL^−1^	0.01–2 pg mL^−1^	Serum	[44]
	GO-PTCNH2	Direct	-	Photoeletrochemical	3.33 pg mL^−1^	10–100 ng mL^−1^	Serum	[73]

LOD, limit of detection, LR, linear range, AP, alkaline phosphatase; AGA, antigliadin Abs; anti-tTG, anti-transglutaminase Abs; CV, cyclic voltammetry; CSF, cerebrospinal fluid; CNC, carbon nanocomposite; DPV, differential pulse voltammetry; GO, graphene oxide; GADA, glutamate decarboxylase Ab; HRP, horseradish peroxidase; HIgG, human immunoglobulin G; IgA, immunoglobulin A; IgB, immunoglobulin B; IL-17, interleukin 17; IL-12, interleukin 12; ITO, indium tin oxide; MIF, Macrophage migration inhibitory factor; MBP, myelin basic protein; NTiP-titanium nanoparticles; PDA, polydopamine; POD, peroxidase; PTCNH_2_, amino-terminated perylene derivative; SWV, square wave voltammetry; SPE, screen printed electrode; TNFα, tumor necrosis factor alpha; Thi, thionine.

**Table 2 biosensors-09-00038-t002:** Peptide based biosensors developed in recent years.

Analyte	Electrode Architecture	Method	Peptide Sequences	Label	LD (ng mL^−1^)	LOQ (ng mL^−1^)	Linear Range (ng mL^−1^)	Real Samples	Ref.
MMP-14	Gold electrode	DPV	VMDGYPMP-(CH_2_)_6_-Cys	CIS-Fc	3 10^−4^	10^−3^	10^−3^–10^−2^	-	[78]
MMP-14		EIS	Cys- (CH_2_)_6_—VMDGYPMP-NH-CO-Fe	-	0.03	0.1	0.1–7	-	
Aβ1 Ab	SPE	CV	DAEFRHDSGYEVHHQKLVFFAEDVGSNKGAI IGLMVGGVV (Aβ1-40)	-			0–10	-	[79]
MMP-7	Au-rGO/MB-SA +PdNP	SWV	NH2-KKKRPLALWRSCCC-SH	-	3 10^−6^	10^−5^	10^−5^–10	Spiked serum samples	[80]
EGFR	Gold electrode	DPV	YHWYGYT- PQNVI	9-mercapto-1-nonanol	3.7 10^−5^	10^−4^	10–10^−4^	Diluted human serum	[81]
Type IV collagenase	QCM gold electrode	QCM	AuNP modified P	-	0.96	10	10–60	Spiked serum samples	[82]
Type IV collagenase		QCM	P	-	21	40	40–120	-	
JIA—IgG	SPE	DPV	ACSSWLPRGCGGGS	-	1:300 diluted serum		1:10–1:300 diluted serum	Real patients serum	[83]
MMP-9	Gold SPE	EIS	Leu–Gly–Arg–Met–Gly–Leu–Pro–Gly–Lys	Dextran		50	50–400	-	[84]
MMP-9	Gold electrode	SWV	Gly-Pro-Leu-Gly-Met-Trp-Ser-Arg-Cys	MB	6 10^−2^ nM		6 10^−2^–50 nM	Spiked serum samples	[77]
MMP-7	AuNP-GCE- P-PtNPs-S_1_	DPV	NH2-KKKRPLALWRSCCC-SH	-	0.05 10^−3^	2 10^−3^	2 10^−3^–20	-	[76]

LD – Limit of detection; LOQ – Limit of quantification; DPV—Differential Pulse Voltammetry; EIS—Electrochemical Impedance Spectroscopy; CV—Cyclic Voltammetry; SWV—Square Wave Voltammetry; QCM—Quartz Crystal Microbalance; SPE—Screen Printed Electrode; GCE—Glassy Carbon Electrode; CIS-Fc—ferrocene carboxylic acid; MMP—Matrix Metalloproteinase; Aβ1 Ab—amyloid-β1 Antibody; Au-rGO/MB-SA—reduced graphene oxide-Au/methylene blue-sodium alginate hydrogel; PdNP—Pd Nanoparticles; EGFR—Epidermal growth factor receptor; MB—methylene blue; AuNP—Au Nanoparticles; JIA—Juvenile idiopathic arthritis; IgG—Immunoglobulin G; PtNP—Pt Nanoparticles; S_1_—single stranded DNA; P—Peptide.

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
