# Peer review of "Electrochemical Biosensors as Potential Diagnostic Devices for Autoimmune Diseases"

_biosensors, 2019, doi:10.3390/bios9010038_

Round 1

Reviewer 1 Report

Review: Electrochemical Immunosensors for Autoimmune Disease”

Manuscript ID: biosensors-414131

Recommendation: Reject

The paper does not provide sufficient information, point of view or knowledge to warrant publication. It does not address the issues of electrochemical sensors, biomarkers and autoimmune disease in a useful and methodical manner. The manuscript is written so poorly with inconsistent terminology, it is very hard to follow and understand. The manuscript does not provide a reader coming new to the topics with a helpful perspective. For a researcher knowledgeable in these areas, the paper does not provide any insights. References and titles relevant to the topics could be obtained directly via a literature search.

The manuscript contains many (too many) spelling mistakes, incomplete sentences, tautology, misused words, non sequiturs, undefined abbreviations. The manuscript reads as if it was not proofread. There is too little useful information on which to suggest major revision. As such, this reviewer recommends the manuscript be rejected.

Some specific comments:

1.    The terminology can be difficult to follow, namely in the use of “immun_” i.e., immunosensor, immunoassay (ELISA), autoimmune, immune system.

2.    The title is misleading. Immunosensors use antibodies as bioreceptors. The paper describes the use of Abs, Apts, microRNA, and peptides as immunosensors. The title should be “electrochemical biosensors” to properly reflect the content.

3.    The abstract is poorly written and totally confusing. This reviewer wonders the analytical chemistry level of understanding of the authors. The abstract mentioned “microfluidics” but it was not discussed later in the document.

4.    In the introduction, line 72, they provided the IUPAC definition of “biosensor” without citing a reference material, and went off the definition throughout the document. They used “biosensor” and “immunosensor” randomly and inconsistently. 

5.    On line 77, the authors stated “The generated signal is directly or indirectly proportional with the analyte concentration.” This statement is false for a number of reasons (too long to explain all). Primarily it is incorrect from the perspective of receptor-ligand binding interaction. Either the authors do not understand the meaning of “proportional” or they do not understand receptor-ligand binding. “Proportional” implies that the plot of signal vs concentration of ligand (analyte) is linear. The plot of such is NOT linear. It is NOT proportional. The mathematical expression for receptor-ligand binding is derived from the law of mass action and formulated as the Langmuir binding isotherm. For a biosensor, when signal is plotted versus ligand (analyte) concentration, the result is a hyperbola with signal = 0 for [ligand] = 0, and signal = maximum (and unchanging) value at high concentration of ligand. In other words, the sensitivity of a biosensor continuously decreases from maximum value at low concentration, ie, when [ligand] approaches zero, to a sensitivity of zero at high concentration. The generated signal is NOT proportional to analyte. Proportional means that that sensitivity, ie the slope (delta signal/delta concentration) is constant. Not so.

6.    This reviewer don’t see the point of Sections 3 “immobilization methods and platforms” in relation to this review paper. It is too general and doesn’t give a take home information with respect of autoimmune disease detections by electrochemistry. They should have focused reviewing what has been done in developing electrochemical biosensors for autoimmune biomarkers detection.

7.    Like point 6, Section 4.1 paragraphs 2 and 3 have nothing to do with topic the authors wanted to review. On line 148, they wrongly categorized chronoamperometry with voltammetry, which makes the proficiency of the authors in electrochemistry questionable.

8.    On page 6, lines 249 – 269 have nothing to do with electrochemical biosensors, which is the topic of the review manuscript. The same is true for lines 277 – 291.

9.    The entire discussion on page 10 about the use of peptides as biorecognition elements in biosensors has nothing to do with the topic under review.

10. The cited reference #4 should be 2017, 65, 111-121. Reference #4 is an excellent review paper on immunosensor for autoimmune diseases biomarkers detection, which summarizes work done in all areas of signal transduction (optical, electrochemical and mechanical), and was misquoted by the authors on line 45. In fact, it appears to this reviewer, the authors of the manuscript under consideration borrowed words and statements from reference paper 4, sometimes twisting them (due to poor language) to the point they lost their meaning. 

Author Response

Answer to reviewer 1:

The paper does not provide sufficient information, point of view or knowledge to warrant publication. It does not address the issues of electrochemical sensors, biomarkers and autoimmune disease in a useful and methodical manner. The manuscript is written so poorly with inconsistent terminology, it is very hard to follow and understand. The manuscript does not provide a reader coming new to the topics with a helpful perspective. For a researcher knowledgeable in these areas, the paper does not provide any insights. References and titles relevant to the topics could be obtained directly via a literature search.

The manuscript contains many (too many) spelling mistakes, incomplete sentences, tautology, misused words, non sequiturs, undefined abbreviations. The manuscript reads as if it was not proofread. There is too little useful information on which to suggest major revision. As such, this reviewer recommends the manuscript be rejected.

We would like to thank the reviewer for the useful suggestions which aim to improve the quality of our work.

The entire manuscript was reconsidered, some parts were deleted, and other new parts were added. The language was also corrected by an English native speaker.

Some specific comments:

1.                   The terminology can be difficult to follow, namely in the use of “immun_” i.e., immunosensor, immunoassay (ELISA), autoimmune, immune system.

Where possible, abbreviations were used.

2.    The title is misleading. Immunosensors use antibodies as bioreceptors. The paper describes the use of Abs, Apts, microRNA, and peptides as immunosensors. The title should be “electrochemical biosensors” to properly reflect the content.

The title was changed after the reviewer’s suggestion.

3.    The abstract is poorly written and totally confusing. This reviewer wonders the analytical chemistry level of understanding of the authors. The abstract mentioned “microfluidics” but it was not discussed later in the document.

The abstract was reconsidered and modified in accordance to the content of the manuscript:

4.    In the introduction, line 72, they provided the IUPAC definition of “biosensor” without citing a reference material, and went off the definition throughout the document. They used “biosensor” and “immunosensor” randomly and inconsistently.

According to IUPAC definition the term biosensor means “A device that uses specific biochemical reactions mediated by isolated enzymes, immune-systems, tissues, organelles or whole cells to detect chemical compounds usually by electrical, thermal or optical signals.” [PAC, 1992, 64, 143. Glossary for chemists of terms used in biotechnology (IUPAC Recommendations 1992), doi:10.1351/pac199264010143; https://goldbook.iupac.org/html/B/B00663.html].

In fact, in the technical report and afterward, in the paper published by D. R. Thevenot et al. [Pure Appl. Chem. 71, 2333±2348, Biosensors & Bioelectronics 16 (2001) 121–131] it could be found the use of term biosensors when discussing about analytical devices based on antigen-antibody interaction: “Antibody-antigen interaction. The most developed examples of biosensors using biocomplexing receptors are based on immunochemical reactions, i.e. binding of an antigen (Ag) to a specific antibody (Ab). Formation of such Ab-Ag complexes has to be detected under conditions where non-specific interactions are minimized.”

In the light of this information, the authors considered that for readability reasons, it was proper to use the term biosensors when referring to immunosensors.

5.    On line 77, the authors stated “The generated signal is directly or indirectly proportional with the analyte concentration.” This statement is false for a number of reasons (too long to explain all). Primarily it is incorrect from the perspective of receptor-ligand binding interaction. Either the authors do not understand the meaning of “proportional” or they do not understand receptor-ligand binding. “Proportional” implies that the plot of signal vs concentration of ligand (analyte) is linear. The plot of such is NOT linear. It is NOT proportional. The mathematical expression for receptor-ligand binding is derived from the law of mass action and formulated as the Langmuir binding isotherm. For a biosensor, when signal is plotted versus ligand (analyte) concentration, the result is a hyperbola with signal = 0 for [ligand] = 0, and signal = maximum (and unchanging) value at high concentration of ligand. In other words, the sensitivity of a biosensor continuously decreases from maximum value at low concentration, ie, when [ligand] approaches zero, to a sensitivity of zero at high concentration. The generated signal is NOT proportional to analyte. Proportional means that that sensitivity, ie the slope (delta signal/delta concentration) is constant. Not so.

We thank the reviewer for this suggestion. In order to avoid any confusion this sentence was deleted.

6.    This reviewer don’t see the point of Sections 3 “immobilization methods and platforms” in relation to this review paper. It is too general and doesn’t give a take home information with respect of autoimmune disease detections by electrochemistry. They should have focused reviewing what has been done in developing electrochemical biosensors for autoimmune biomarkers detection.

All the paragraphs were renamed and the information was re-organized in order to become relevant for the readers.

7.    Like point 6, Section 4.1 paragraphs 2 and 3 have nothing to do with topic the authors wanted to review. On line 148, they wrongly categorized chronoamperometry with voltammetry, which makes the proficiency of the authors in electrochemistry questionable.

We believe that is a misunderstanding. In section 4.1. paragraphs 2 and 3 was written “Voltammetry (CV, DPW, SWV) and chronoamperometry are the electrochemical methods most commonly used to convert the Ab-Ag binding event into a measurable signal, the current response.” Several examples of electrochemical methods employed for immunosensors characterization were listed. We agree that we are talking about two different types of methods. We did not intend to categorize electrochemical methods just to give a few examples of the most employed methods.

8.    On page 6, lines 249 – 269 have nothing to do with electrochemical biosensors, which is the topic of the review manuscript. The same is true for lines 277 – 291.

The subtitles were replaced and the whole manuscript was reorganised in order to have a broad look over electrochemical methods, biosensor’ design, type of AD biomarkers and immobilization strategies of the bioelements.

9.    The entire discussion on page 10 about the use of peptides as biorecognition elements in biosensors has nothing to do with the topic under review.

The abstract and the introduction were modified accordingly in order to have the possibility to presents the latest achievement in the field of electrochemical biosensors for autoimmune diseases in which beside antibodies and aptamers in the last years are used also the peptides.

10. The cited reference #4 should be 2017, 65, 111-121. Reference #4 is an excellent review paper on immunosensor for autoimmune diseases biomarkers detection, which summarizes work done in all areas of signal transduction (optical, electrochemical and mechanical), and was misquoted by the authors on line 45. In fact, it appears to this reviewer, the authors of the manuscript under consideration borrowed words and statements from reference paper 4, sometimes twisting them (due to poor language) to the point they lost their meaning.

We agree with the reviewer about the reference 4 and we inserted the correct citation. We only want to underline that we are talking about a review type manuscript, which gather information from different sources (all cited) and reinterpret the general or specific information through our point of view. We tried to presents new approaches published in the last 5 years and undoubtly, some terms and examples could be found in both reviews. Hopefully, after the revision of the manuscript and the answers to all reviewers’ observation, this reviewer will have a better impression about this manuscript.

Reviewer 2 Report

This manuscript summerised the immunosensors and its applications to daignosis for autoimmune diseases. Authors explained the structure, and priciples of immunosensors well. The topic of this manuscript is well fit to the Biosensors and organized well with good readability. However, authors described "normal" immunosensors and there is lack of explanation about the detection stratages for the autoimmune diseases. Normally, biomarkers of autoimmune diseases are antibody and the detection stratage of antibody is different from the detection of antigen. If authors modify follow things, it would be better.

Figure 1 is the schematic diagram of immunosensor for the detection of "antigen". For the good readability, it would be better to add other figure about the detection of "antibody" for the diagnosis of autoimmune disease.

Page 4, line 170. authors describes about the electroactive labeling of immunosensors for ADs. However, referneces 31, 32 is not related to the detection of autoimmune disease biomarker.

Authors should add the description of the detection stratage of antibody for the daignosis of autoimmune diseases.

This manuscript includes only one figure. It would be helpful if authors supply 3~4 more figures.

Author Response

Comments and Suggestions for Authors

This manuscript summerised the immunosensors and its applications to daignosis for autoimmune diseases. Authors explained the structure, and priciples of immunosensors well. The topic of this manuscript is well fit to the Biosensors and organized well with good readability. However, authors described "normal" immunosensors and there is lack of explanation about the detection stratages for the autoimmune diseases. Normally, biomarkers of autoimmune diseases are antibody and the detection stratage of antibody is different from the detection of antigen. If authors modify follow things, it would be better.

We thank the reviewer for the useful suggestion which will improve the quality of the manuscript.

Regarding the different strategies for antibodies and antigens detection, several sentences were added as well as the figure 1 which was reconsidered.

Figure 1 is the schematic diagram of immunosensor for the detection of "antigen". For the good readability, it would be better to add other figure about the detection of "antibody" for the diagnosis of autoimmune disease.

We thank the reviewer for this suggestion Figure 1 B was added presenting the general detection strategy for antibodies.

Page 4, line 170. authors describes about the electroactive labeling of immunosensors for ADs. However, referneces 31, 32 is not related to the detection of autoimmune disease biomarker.

Although the references were not related to the detection of an autoimmune disease biomarker, the authors believe that the strategies used in those papers to immobilize bioelements specific for other target analytes, are simple and advantageous, and can be also employed for biomarkers present in autoimmune disease (for example for the immobilization of Ab for interleukins or HIgG). To improve the clarity, we have modified that paragraph as follows:

“Other immobilization strategies maybe considered and applied for ADs immunosensors, given the fact that they were successfully applied for other targets. Although not reported so far for ADs biosensors, magnetic beads functionalized with various groups, such as avidin, protein A or G, can also be used as support for the immobilization of the Ab or the Ag with the advantage of high loading capacity of biomolecules, easy separation and easy washing steps [18]. Another immobilization strategy is based on using aryl diazonium salts to link the Ab directly onto the electrode by electroaddressing [22] or the use of polymer composites to covalently link the capture Ab in an oriented manner [21]. These strategies they could be easily adapted and employed for the immobilization of bioelements specific for ADs biomarkers.”

Authors should add the description of the detection stratage of antibody for the daignosis of autoimmune diseases.

We added the description of the detection strategy of antibodies on page 3, line109 as follows:

“Most of the immunosensors reported for ADs are based on labeled methods and a direct and indirect format. They usually employ an antigen (Ag) as recognition element since the targets in ADs are usually autoantibodies. Antigens (Ags) are firstly immobilized on the electrode surface. Then the analyte is added (the sample containing autoantibodies present in ADs). The Ags selectively recognize and bind the Abs from the sample, and the formation of the Ag-Ab complex can be assessed using secondary Abs, which are usually labeled with an enzyme. Upon the addition of the enzyme´s substrate the product of the enzymatic reaction (electrochemically active) enables the quantitation of the target analyte.”

This manuscript includes only one figure. It would be helpful if authors supply 3~4 more figures.

Two new figures were added in the manuscript.

Reviewer 3 Report

This review summarized the recent advancement of research on electrochemical immunosensors for diagnosis and monitoring of autoimmune diseases. The review is well organized and have a deep insight in peptide sensor. This is a promising field in the future of electrochemical immunosensors.  

Author Response

Answer to reviewer 3

This review summarized the recent advancement of research on electrochemical immunosensors for diagnosis and monitoring of autoimmune diseases. The review is well organized and have a deep insight in peptide sensor. This is a promising field in the future of electrochemical immunosensors. 

We thank the reviewer for the positive feedback about our manuscript.

Reviewer 4 Report

The manuscript described the state of the art of immunosensors for autoimmune diseases, especially driven to electrochemical detection with a certain emphasis on the use of peptides. In general terms, the review is well written and presents the information with an appropriate order. However, the manuscript lacks on a certain of details that must be solved or changed before publishing. Small details are listed but further comments are related at the end of the comments

-English in general is adequate, but there are some typos as the word “than” rather than “then” on page 10 line 313.

-The use of abbreviations must be consisted, specifically, the abbreviation DAs was early introduced in the manuscript but then sometimes used and others not (e.g. page 2, lines 49, 57, etc).

-The line 62 shall be rephrase because it is unclear

Particularly the manuscript lacks on appropriate references, most of them are not updated for a state-of-the-art review. The conclusions of the review must be complemented and reveal interesting information regarding the biosensors. At the present state of the manuscript the information related is a good compilation of information but lacks on driving to the reader into a new scope or frontier in the topic, which is the main purpose of a review.

Author Response

Answer for reviewer 4

The manuscript described the state of the art of immunosensors for autoimmune diseases, especially driven to electrochemical detection with a certain emphasis on the use of peptides. In general terms, the review is well written and presents the information with an appropriate order.

We would like to thank to the reviewer for recognizing the quality of our study and for the useful suggestions which aim to improve the aspect of the work.

However, the manuscript lacks on a certain of details that must be solved or changed before publishing. Small details are listed but further comments are related at the end of the comments

-English in general is adequate, but there are some typos as the word “than” rather than “then” on page 10 line 313.

We thank the reviewer for the comment; the manuscript has been revised for typos.

Regarding `than` versus `then`, there is the following statement on page 10 “Anti-CPP is more specific than RF and is produced in the mucosal tissues and at the point of inflammation”. Since the statement is about a comparison, the word `than` is correctly used in this sentence. `Than` refers to comparison, while `then` refers to time.

-The use of abbreviations must be consisted, specifically, the abbreviation DAs was early introduced in the manuscript but then sometimes used and others not (e.g. page 2, lines 49, 57, etc).

We thank the reviewer for the comment, the manuscript has now been revised and the abbreviation properly used.

-The line 62 shall be rephrase because it is unclear

Line 62 has been rephrased as follows: “The early diagnosis of RA is essential to avoid an aggressive treatment and to prevent joint damage and disability, therefore there is an urgent need to diagnose RA as early as possible.”

Particularly the manuscript lacks on appropriate references, most of them are not updated for a state-of-the-art review. The conclusions of the review must be complemented and reveal interesting information regarding the biosensors. At the present state of the manuscript the information related is a good compilation of information but lacks on driving to the reader into a new scope or frontier in the topic, which is the main purpose of a review.

Updated references were added. The abstract, introduction and conclusion were reconsidered highlighting relevant aspect of the electrochemical biosensors for autoimmune diseases. A comparative discussion regarding the importance of nanomaterials over the analytical performances of the peptide based sensors was also inserted into the manuscript.

Round 2

Reviewer 1 Report

Manuscript ID: biosensors-414131

Review on revised version (v2): Electrochemical biosensors as potential diagnostic devices for autoimmune diseases

Recommendation: Major revision

The authors made use of the reviewer’s specific comments with respect to textual issues, however there are large portions of the manuscript that still remain poorly written.

Abstract Line19 

“Their low sensitivity, low cost and the easy integration in point of care devices assuring portability are attracting features which justify the increasing interest in their development.” The authors likely mean “high sensitivity”.

Line 82, “… reported in literature in the last years for the detection of Ads …” Specify the length of time? 3, years, 5 years, 10 years?

The basic explanation of biosensors given in section 2 is 30 years old and needs to be shortened.

Line 143, “Additionally, fouling of the electrode surface may occur”. This statement is wrong. The authors say electrode surface fouling is caused by phyisorptoin of the biorecongition element. This is not true. Fouling is caused by non-specific adsorption of interfering proteins in the sample matrix.

Line 149, again here, specify the length of time.

Lines 151-158 make mention of nanomaterials. Good but not very much depth.
Line 155: “The roughened surface generated by the modification of the electrode with nanomaterials facilitates the attachment of a higher number of biomolecules to the electrode surface leading to higher sensitivity.” This is a very important issue in current sensors research. Is there a reference for this statement? Is there actual data? Is there a mathematical expression that relates mass of immobilized receptors to sensor signal? Such would be value added.

Line 187-188, “Usually a labelled secondary Ab ….” Needs explanation.

Line 209 – 211, In Das et al work, what AD was the target disease or target biomarker?

Line 233 – 235, the general statement that EIS does not require redox process is not correct. EIS can be Faradiac (which needs redox probe in solution) or non-Faradiac (doesn’t require redox probe).

Line 242, “sensibility, …” Do they mean “sensitivity,…”?

Line 242, “Sensibility, the electrodes surface were modified with Pt, Au or ITO, …” Did the authors use nanoparticles, nanowires or nanorods of the above metals and metal oxide?

Line 263, “0.03% in the 1970s to 1% to this day” what do the percentages mean? IS it world population?

The authors added more text for section 4 “biosensors based on peptides”. The idea was good; this area is still novel. However, there is no depth to the section. The authors tell us that peptides have 1D, 2D, 3D structures and that peptides can be 10 AA residues in length. We know this. It does not require a review article. Nor does the number of papers published annually between 2008 - 2018 require such. All in all, lines 326 – 352 need to be significantly shortened as most of the discussion has nothing to do with the topic of the review paper. The latter part of the section is fact-oriented, viz., “calibration for anti-CSF114 Abs was 1.25-30 mg/ml … characterized by CV.”

Line 357, define CSF114?

Line 361 compared the result obtained in reference 57 to gold electrode surface used in 59. What was used in reference 57? What was the LOD in 57?

Line 365 – 367, what are the analytical figures of merits of the work in reference 60?

Line 368, define MMPs and their significance.

Line 371, “the best limit of detection was obtained or MMP-7….” needs a reference.

The cartoon in Fig 3 isn’t particularly useful. It shows that they immobilized the PtNP-constructs on the surface, added scissors (the MMP-7 enzyme), and then the DPV sweep changed, although I do not think that the authors defined DPV (differential pulse voltammetry) in the text. The linear range was 2*10-4 – 20 ng ml-1. Similarly, Fig 4 does not help the reader’s interest or knowledge. It contains a slightly different cartoon of an electrode with immobilized structures, add scissors (MMP-9), then current goes down. Calibration gives linear range 0.06-50 nM. So what? What is the relationship between the two techniques, i.e., between 2*10-4 ng and 0.06 nM? Obviously, the relationship would be based on the respective molecular weights of the scissors. In the least the authors could have compared the linear range, they would have noted that MMP-9 reported a greater linear range. Instead of copying the LODs reported in reference 77 and 78 with varying units, the authors could at least convert them and report using the same unit to help the reader understand better the two techniques. Keep space between values and units.

Lines 394 – 397 has nothing to add to the value of the review paper. It feels simply is a blurb to fill up pages. Delete it.

Author Response

The authors made use of the reviewer’s specific comments with respect to textual issues, however there are large portions of the manuscript that still remain poorly written.

We thank the reviewer for its observations and suggestions.

Abstract Line19

“Their low sensitivity, low cost and the easy integration in point of care devices assuring portability are attracting features which justify the increasing interest in their development.” The authors likely mean “high sensitivity”.

We thank the reviewer for this comment. The authors corrected the sentence and replaced “low sensitivity” with “high sensitivity”.

Line 82, “… reported in literature in the last years for the detection of Ads …” Specify the length of time? 3, years, 5 years, 10 years?

We thank the reviewer for the comment. The authors mentioned 5 years.

The basic explanation of biosensors given in section 2 is 30 years old and needs to be shortened.

The explanation was shortened as reviewer suggested.

Line 143, “Additionally, fouling of the electrode surface may occur”. This statement is wrong. The authors say electrode surface fouling is caused by phyisorptoin of the biorecongition element. This is not true. Fouling is caused by non-specific adsorption of interfering proteins in the sample matrix.

The reviewer is right but in our opinion immobilizing the antigen (could be a protein from the matrix) or the antibody (both are not conductive) will create an insulating layer and a barrier for the charge transfer, therefore fouling.

Line 149, again here, specify the length of time.

We do not consider relevant to mention the length of time in the paragraph below. Progress has been made recently and is continuing to be made.

Lines 151-158 make mention of nanomaterials. Good but not very much depth.

The authors wanted only to underline the enhancement of the analytical parameters of the biosensors.

Line 155: “The roughened surface generated by the modification of the electrode with nanomaterials facilitates the attachment of a higher number of biomolecules to the electrode surface leading to higher sensitivity.” This is a very important issue in current sensors research. Is there a reference for this statement? Is there actual data? Is there a mathematical expression that relates mass of immobilized receptors to sensor signal? Such would be value added.

Reference 19. M. Holzinger et al. Front. Chem. 2014, has been added for this statement. This is a very comprehensive review of nanomaterials applied for biosensors development and is mentioned from the beginning of the introduction that “One general advantage of all nanomaterials is the high specific surface thus already enabling the immobilization of an enhanced amount of bioreceptor units”.

Line 187-188, “Usually a labelled secondary Ab ….” Needs explanation.

A concise explanation has been added.

Usually a labeled secondary Ab is used to generate the signal, in order to avoid labeling Abs for each specific Ag. This is a generic labeled IgA/G antiAb.

Line 209 – 211, In Das et al work, what AD was the target disease or target biomarker?

The target was PSA (prostate specific antigen). Although PSA is not an AD marker, the authors believed that this is an important reference to show the usefulness of nanocatalysts used as labels in immunosensors in signal amplification and noise reduction. The only difference being the target antigen this strategy could be easily adapted and apply to other biomarkers including AD.

Line 233 – 235, the general statement that EIS does not require redox process is not correct. EIS can be Faradiac (which needs redox probe in solution) or non-Faradiac (doesn’t require redox probe).

The reviewer is right. We have now corrected the sentence: “EIS is an electrochemical technique widely used as detection method for label-free immunosensors”.

 Line 242, “sensibility, …” Do they mean “sensitivity,…”?

Sensibility has been replaced by sensitivity.

Line 242, “Sensibility, the electrodes surface were modified with Pt, Au or ITO, …” Did the authors use nanoparticles, nanowires or nanorods of the above metals and metal oxide?

The line was modified according to reviewer observation “In order to obtain a good sensitivity, the electrodes has been modified with materials like platinum, gold, TiO2 or polymers.”

Line 263, “0.03% in the 1970s to 1% to this day” what do the percentages mean? IS it world population?

It is world population. The sentence was modified as follows “The worldwide prevalence of celiac disease increased from 0.03% in the 1970s to 1% to this day”.

The authors added more text for section 4 “biosensors based on peptides”. The idea was good; this area is still novel. However, there is no depth to the section. The authors tell us that peptides have 1D, 2D, 3D structures and that peptides can be 10 AA residues in length. We know this. It does not require a review article. Nor does the number of papers published annually between 2008 - 2018 require such. All in all, lines 326 – 352 need to be significantly shortened as most of the discussion has nothing to do with the topic of the review paper. The latter part of the section is fact-oriented, viz., “calibration for anti-CSF114 Abs was 1.25-30 mg/ml … characterized by CV.”

The entire paragraph related to peptides structures was reconsidered and shorten as the reviewer suggested. By presenting the number of the publications, the authors wanted to underline the increasing interest toward the use of peptides in the last ten years.

Line 357, define CSF114?

CSF114 was defined.

Line 361 compared the result obtained in reference 57 to gold electrode surface used in 59. What was used in reference 57? What was the LOD in 57?

In reference 57 the same gold surface was used. The LOD was not reported in the article.

Line 365 – 367, what are the analytical figures of merits of the work in reference 60?

The analytical performances were added as reported in ref. 60.

Line 368, define MMPs and their significance.

MMPs were defined in text.

Line 371, “the best limit of detection was obtained or MMP-7….” needs a reference.

The reference was added.

The cartoon in Fig 3 isn’t particularly useful. It shows that they immobilized the PtNP-constructs on the surface, added scissors (the MMP-7 enzyme), and then the DPV sweep changed, although I do not think that the authors defined DPV (differential pulse voltammetry) in the text. The linear range was 2*10-4 – 20 ng ml-1. Similarly, Fig 4 does not help the reader’s interest or knowledge. It contains a slightly different cartoon of an electrode with immobilized structures, add scissors (MMP-9), then current goes down. Calibration gives linear range 0.06-50 nM. So what? What is the relationship between the two techniques, i.e., between 2*10-4 ng and 0.06 nM? Obviously, the relationship would be based on the respective molecular weights of the scissors. In the least the authors could have compared the linear range, they would have noted that MMP-9 reported a greater linear range. Instead of copying the LODs reported in reference 77 and 78 with varying units, the authors could at least convert them and report using the same unit to help the reader understand better the two techniques. Keep space between values and units.

The LOD were reported with the same unit in order to be easy comparable with the results from ref. 77 and 78.

Lines 394 – 397 has nothing to add to the value of the review paper. It feels simply is a blurb to fill up pages. Delete it.

The lines have been deleted.

Reviewer 4 Report

Authors answered the different requests from the reviewers. The manuscript has improved and it is more updated.

Conclusions have been complemented as well as references. 

Author Response

Reviewer 4

Authors answered the different requests from the reviewers. The manuscript has improved and it is more updated.

Conclusions have been complemented as well as references.

We thank the reviewer for useful comments.